# Hepatic *Nfe2l2* Is Not an Essential Mediator of the Metabolic Phenotype Produced by Dietary Methionine Restriction

**DOI:** 10.3390/nu13061788

**Published:** 2021-05-24

**Authors:** Han Fang, Kirsten P. Stone, Sujoy Ghosh, Laura A. Forney, Landon C. Sims, LeighAnn Vincik, Thomas W. Gettys

**Affiliations:** 1Laboratory of Nutrient Sensing & Adipocyte Signaling, 6400 Perkins Road, Pennington Biomedical Research Center, Baton Rouge, LA 70808, USA; han.fang@pbrc.edu (H.F.); kirsten.stone@pbrc.edu (K.P.S.); lsims@ucdavis.edu (L.C.S.); Lvincikk@gmail.com (L.V.); 2Laboratory of Computational Biology, Pennington Biomedical Research Center, Baton Rouge, LA 70808, USA; sg45653@gmail.com; 3Program in Cardiovascular and Metabolic Disorders and Center for Computational Biology, Duke-NUS Medical School, Singapore 169857, Singapore; 4Department of Integrative Biology and Pharmacology, University of Texas Health Science Center at Houston, 7000 Fannin St, Houston, TX 77030, USA; Laura.A.Bobart@uth.tmc.edu

**Keywords:** essential amino acid, nutrient sensing, obesity, integrated stress response, Nfe2l2, FGF21

## Abstract

The principal sensing of dietary methionine restriction (MR) occurs in the liver, where it activates multiple transcriptional programs that mediate various biological components of the response. Hepatic *Fgf21* is a key target and essential endocrine mediator of the metabolic phenotype produced by dietary MR. The transcription factor, *Nfe2l2*, is also activated by MR and functions in tandem with hepatic *Atf4* to transactivate multiple, antioxidative components of the integrated stress response. However, it is unclear whether the transcriptional responses linked to *Nfe2l2* activation by dietary MR are essential to the biological efficacy of the diet. Using mice with liver-specific deletion of *Nfe2l2* (*Nfe2l2*^fl/(Alb)^) and their floxed littermates (*Nfe2l2*^fl/fl^) fed either Control or MR diets, the absence of hepatic *Nfe2l2* had no effect on the ability of the MR diet to increase FGF21, reduce body weight and adiposity, and increase energy expenditure. Moreover, the primary elements of the hepatic transcriptome were similarly affected by MR in both genotypes, with the only major differences occurring in induction of the P450-associated drug metabolism pathway and the pentose glucuronate interconversion pathway. The biological significance of these pathways is uncertain but we conclude that hepatic *Nfe2l2* is not essential in mediating the metabolic effects of dietary MR.

## 1. Introduction

Reducing dietary methionine concentration from ~0.8% to 0.17% is the basis for the well-studied model of dietary methionine restriction (MR) first described by Orentreich et al. [1]. In addition to extending lifespan in rodents [1,2,3], dietary MR also produces short-term metabolic benefits that include inhibition of weight gain, reduced fat accumulation, reductions in tissue and circulating lipid levels, and enhancement of overall insulin sensitivity [4,5,6,7]. Dietary MR limits body weight and fat accumulation by increasing energy expenditure (EE) [5,8], and many of the accompanying metabolic improvements produced by MR are secondary to reduced adiposity and the associated decrease in markers of inflammation [9,10]. However, a critically important response to sensing of low methionine is the induction of hepatic fibroblast growth factor 21 (FGF21) [7,11], as most metabolic effects of dietary MR are FGF21-dependent [12]. The MR-induced increase in EE is due to FGF21 acting in the brain to increase sympathetic outflow to adipose tissue [13,14], but FGF21 also enhances insulin sensitivity through direct effects in peripheral tissues [12]. In addition, dietary MR produces significant inhibition of hepatic lipid metabolism through transcriptional effects that are FGF21 independent [12]. An important unresolved question is how the hepatic sensing of reduced methionine intake is translated into the complex series of transcriptional programs that are initiated within hours of introduction of the MR diet.

Transcriptional activation of FGF21 is an essential component of the integrated stress response (ISR) that occurs in the liver after dietary restriction of protein or essential amino acids like methionine [7,15]. The initial step of the ISR involves phosphorylation of eukaryotic initiation factor eIF2α by serine/threonine kinases that are activated by different cellular stresses (e.g., general control nonderepressible 2 (GCN2), heme-regulated inhibitor (HRI), PKR-like endoplasmic reticulum kinase (PERK) and protein kinase double-stranded RNA dependent (PKR). Although each is capable of phosphorylating and activating eIF2α, GCN2 and PERK are the kinases most responsive to nutritional perturbations. Activation of eIF2α produces an overall decrease in protein synthesis by limiting ribosomal translation of mRNA [16,17,18], while selectively de-repressing translation of genes containing specific upstream open reading frames in their promoters. Through increased duration of ribosomal scanning and re-initiation efficiency [19], proteins such as activating transcription factor 4 (ATF4) are recruited to promoters of genes such as FGF21 that mediate cellular stress responses [20,21]. It was originally thought that MR-induced stress signaling occurred solely through the canonical GCN2–eIF2α–ATF4 pathway [20,22,23,24], but recent work has shown that MR can produce its full range of metabolic effects in mice lacking GCN2 [25]. PERK was suggested as an alternative pathway linking MR to eIF2α in the absence of GCN2 [25], but subsequent studies showed that liver-specific deletion of PERK did not compromise the ability of MR to reduce body weight or increase eIF2α phosphorylation [26]. Another more recent study showed that liver-specific deletion of ATF4 also failed to compromise the ability of MR to increase transcription of hepatic FGF21 or reduce body weight [27]. Collectively, these studies suggest that each of these molecules are dispensable and not required for dietary MR to either activate the ISR or produce its full range of metabolic effects.

Nuclear factor erythroid 2-related factor 2 (Nfe2l2, NRF2) is an important stress-sensitive transcription factor that functions in concert with ATF4 to regulate components of the ISR program [28,29]. We have proposed that dietary MR also signals through PERK in the liver [25] to activate Nfe2l2 and its transcriptional program [30,31]. Nfe2l2 is a master regulator of oxidative stress responses in the liver [32], where its target genes catalyze conjugation of glutathione (GSH), amino acids, sulphates, glucuronic acid, acetyl residues, and methyl residues to activated xenobiotics. These reactions prevent ROS generation by xenobiotic auto-oxidation and irreversible protein inactivation by oxidative stress (reviewed in [33]). Given that dietary MR produces a significant decrease in hepatic GSH that could be activating Nfe2l2 through PERK [30,31,34,35,36,37,38,39], we have proposed that Nfe2l2 could be an important mediator of the transcriptional and biological responses to dietary MR [25]. Using mice with liver-specific deletion of *Nfe2l2* fed MR diets, it is shown that the absence of Nfe2l2 muted the diet-induced induction of P450-associated drug metabolism pathways, some Nfe2l2 target genes, and pentose glucuronate interconversion pathways, but had no significant effect on the ability of dietary MR to produce its primary transcriptional effects in liver or any of its metabolic effects.

## 2. Materials and Methods

### 2.1. Animals and Diets

All experiments were reviewed and approved by the Institutional Animal Care and Use Committee at Pennington Biomedical Research Center using the guidelines established by the National Research Council, Animal Welfare Act, and Public Health Service Policy on the humane care and use of animals. *Nfe2l2* floxed (C57BL/6-Nfe2l2tm1.1Sred/SbisJ; Stock No: 025433, *Nfe2l2^fl/fl^*) and Albumin-cre (B6.Cg-Speer6-ps1Tg(*Alb-cre*)21Mgn/J; Stock No: 003574) mice were obtained from Jackson Laboratory (Ellsworth, ME). Homozygous *Nfe2l2^fl/fl^* mice were bred to mice homozygous for the floxed *Nfe2l2* and with one copy of the *Alb-cre* gene, resulting in litters consisting of 50% *Alb-cre*-positive homozygous *Nfe2l2* floxed mice (*Nfe2l2^fl/(Alb)^*). The homozygous *Nfe2l2* floxed littermates (*Nfe2l2^fl/fl^*) served as control animals. The genotype of all mice was confirmed by real time PCR to confirm the absence of hepatic *Nfe2l2* mRNA. Animals were fed Purina diet #5001 from weaning until six weeks of age, at which time they were singly housed in shoebox cages with corncob bedding. All mice were switched to the previously described Control diet (Con; Dyets, Inc., Bethlehem, PA, USA) at this time and provided the Con diet for the subsequent two weeks before randomization to either the Con or MR diet groups. The Con diet contained 0.86% methionine and no cysteine, whereas the MR diet contained 0.17% methionine and no cysteine [5]. Diets were formulated as extruded pellets and fed ad libitum. The energy content of both diets was 15.96 kJ/g, with 18.9% of energy from fat, 64.9% from carbohydrates, and 14.8% from a custom L-amino acid mixture. Housing temperature was maintained at 23 °C and lights were on a 12 h on/off schedule. Water was provided ad libitum, and food and water intake were measured at weekly intervals. Additionally, body weight and composition were measured weekly. Body composition was assessed using nuclear magnetic resonance (NMR) spectroscopy (Bruker Minispec, Billerica, MA, USA). At the end of the study, animals were fasted for four hours prior to euthanasia via CO_2_-induced narcosis and decapitation. Trunk blood was collected for serum analyses and livers were harvested and snap frozen in liquid nitrogen and stored at −80 °C until analysis.

### 2.2. Experiment 1

Six-week-old male *Nfe2l2^fl/fl^* (Wild Type, WT) or *Nfe2l2^fl/(Alb)^* (liver-specific *NRF2* Knockout, KO) mice (*n* = 16–20 per genotype) were housed at 22–23 °C and given the Con diet for two weeks prior to being adapted to the Promethion indirect calorimetry (IDC) system (Sable Systems, Las Vegas, NV, USA) for three days. On day 3, half the mice of each genotype were randomized to receive the MR diet, while the remaining half continued to receive the Con diet. Mice remained in the calorimeter for an additional nine days to measure how energy expenditure (EE) changed over time after initial exposure to dietary MR.

### 2.3. Experiment 2

Six-week-old male WT or *NRF2* KO mice (*n* = 16–20 per genotype) were housed at 22–23 °C and given the Con diet for two weeks prior to being randomized to receive Con or MR diet for the following eight weeks. Body weight, composition, and food intake measurements were made at weekly intervals as described above. Following eight weeks on their respective diets, animals were acclimated for one week to the Promethion IDC system, followed by five days of data collection for assessment of EE. Animals were then returned to their home cages for a two-week re-equilibration prior to euthanasia and tissue harvest.

### 2.4. Analysis of Energy Expenditure

VO_2_ is expressed as liters (L) of O_2_ consumed per h, while Respiratory Exchange Ratio (RER) is the ratio of VCO_2_ produced to VO_2_ consumed. EE was calculated as (VO_2_ × (3.815 + (1.232 × RER)) × 0.96 kCal/h) × 4.019 kJ/kCal, and expressed as kJ/h/mouse as described by the manufacturer (Promethion, Sable Systems, N Las Vegas, NV, USA). Group differences in 24 h EE (kJ/h/mouse) at study’s end were compared using Analysis of Covariance (ANCOVA) (JMP Software, Version 15; SAS Institute Inc., Cary, NC, USA) to calculate least squares means that accounted for variation in EE attributable to differences in lean mass, fat mass, and activity among the mice. To evaluate the time-dependent changes in EE at the beginning of *Experiment 1*, the initial diet-dependent change in EE was calculated by subtracting basal EE from the daily nighttime measures of EE during the nine days after initial exposure to the MR diet on day 0. The MR diet-induced changes in mean nighttime EE measurements were compared to the corresponding measures of nighttime EE in mice of each genotype on the Con diet using a repeated measures two-way ANOVA and the Bonferroni correction to account for multiple comparisons as before [13].

### 2.5. RNA Isolation and qPCR of Nfe2l2 and ER Stress Target Genes

Total RNA was isolated from livers of WT and *NRF2* KO mice fed either Con or MR diets using RNeasy Mini kit (Qiagen, Valencia, CA, USA). RNA concentrations were measured using a Nanodrop ND-1000 spectrophotometer (Nanodrop Technologies, Wilmington, DE, USA). Total RNA was analyzed using the Agilent Bioanalyzer RNA 1000 chip as a quality control step to confirm the integrity of the RNA. Samples were verified to have RIN values > 7, indicating high quality RNA. Two μg of total RNA was used for reverse transcription to produce cDNA. mRNA expression for previously identified *Nfe2l2-* and ER stress-target genes [25] was measured using 10 ng cDNA via quantitative PCR using SYBR Green (Bio-Rad, Valencia, CA, USA), and mRNA concentrations of each target gene were standardized to cyclophilin expression. Primer sequences are provided in Appendix A.

### 2.6. RNAseq Analysis

RNA samples were processed for library construction using the Lexogen Quant-Seq 3’ mRNA-Seq Library Prep Kit. Completed libraries were analyzed on the Agilent Bioanalyzer High Sensitivity DNA chip to verify correct library size. All libraries were pooled in equimolar amounts and sequenced on the Illumina NextSeq 500 at 75 bp forward read and 6 bp forward index read. Primary data analysis was performed using the Lexogen Quantseq pipeline 2.3.6 FWD on the Bluebee platform for quality control, mapping, and read count tables. The gene expression profiles were assessed from 6 replicates for each genotype of each dietary group. CLC Genomics Workbench was used to process data. The expression profiling data have been deposited in NCBI under GEO accession GSE162964.

### 2.7. Bioinformatics Analysis

Prior to differential gene expression analysis, scaled normalized count data for samples from the 4 treatment groups (WT Con, WT MR, *NRF2* KO Con, and *NRF2* KO MR) were analyzed via principal component analysis (PCA) (using *prcomp* package in R, http://www.R-project.org) to cluster samples based on gene expression similarities, and to identify potential outliers. After removal of two outlier samples (see Appendix A), differential analysis of RNA read count data was performed using DESeq2 software [40], which models read counts as a negative binomial distribution and uses an empirical Bayes shrinkage-based method to estimate signal dispersion and fold changes. Gene expression signals were logarithmically transformed (to base 2) for all downstream analyses (the lowest expression value being set to 1 for this purpose). Genes with an absolute log fold change ≥ 1 and false discovery rate (FDR) of 5% were considered as differentially expressed.

*Ingenuity Pathway analysis (IPA)*—Pathway over-representation analysis was conducted using IPA (QIAGEN Inc., (Valencia, CA, USA, https://www.qiagenbioinformatics.com/products/ingenuity-pathway-analysis)), considering 3179 differentially expressed genes from the WT MR to WT Con samples and 3405 differentially expressed genes from the *NRF2* KO MR to *NRF2* KO Con samples (absolute log fold change ≥ 1.3, FDR ≥ 0.3). Within IPA, the Upstream Regulator Analysis module was utilized to identify putative gene regulators responsible for the observed transcriptional patterns produced by the MR diet compared to the Con diet in the two genotypes. Several possible types of upstream regulators were considered but special emphasis was placed on identifying activation or inhibition of transcription factors, cytokine receptors, G protein-coupled receptors, and ligand-dependent nuclear receptors. Differential regulation of canonical pathways by the MR diet across genotype was also examined. Upstream regulators and canonical pathways with an activation z-score ≥2 or ≤2 were considered to be activated or inhibited, respectively. Heat maps were used to visualize upstream regulators and canonical pathways that were differentially affected by the MR diet in the two genotypes.

*Pre-ranked GSEA analysis*—Enrichment analysis of biological pathways (gene sets) was conducted via gene set enrichment analysis (GSEA) [41]. Specifically, the GSEA pre-ranked option was used [41] and enrichment was computed by first ranking all genes based on their log fold changes in the respective comparisons. Enrichment was computed either on user-defined custom pathways or pathways present in the Kyoto Encyclopedia of Genes and Genomes (KEGG) database [42] available from the Molecular Signatures Database repository (MSigDb, http://software.broadinstitute.org/gsea/msigdb [43]). Statistical significance for the observed enrichment was ascertained by permutation testing over size-matched gene sets. Gene sets with FDR ≤ 5% were considered as significantly enriched [44]. The individual contributions of pathway genes to the pathway enrichment signal were visualized via enrichment plots depicting the trajectory of a normalized pathway enrichment score against the rank of the pathway genes in the context of the full gene list.

### 2.8. Serum Metabolite Analyses

Fasting serum FGF21 (R&D Systems; Minneapolis, MN, USA) was determined via enzyme-linked immunosorbent assays (ELISA) according to the manufacturers’ protocol.

### 2.9. Data Analysis

Body weight, adiposity, food intake, water intake, serum FGF21, and expression of specific genes were analyzed using two-way ANOVA (GraphPad Prism; San Diego, CA, USA) with genotype and diet as main effects, and residual variance used as the error term to calculate standard errors for group comparisons. Group differences in EE (kJ/h/mouse) at the end of the study were compared using ANCOVA as described previously [14,45], while EE changes over time during the run-in period (e.g., *Exp 1*) were tested using repeated measures two-way ANOVA as previously described [13]. The least square means ± SEM for each genotype × diet × time interaction were compared using residual variance. Protection against type I errors for all comparisons was set at 5% (α = 0.05).

## 3. Results

Cre recombinase driven by the albumin promoter (Alb) excises exon 5 in the hepatic *Nfe2l2* gene, resulting in a null allele. Primers were designed to detect mRNA transcribed from exon 5 including the excised portion and used to measure *Nfe2l2* gene expression in the livers of WT (*Nfe2l2^fl/fl^*) and *NRF2* KO (*Nfe2l2^fl/(Alb)^*) mice. Expression of *Nfe2l2* mRNA in livers of *NRF2* KO mice was decreased by 70–80% compared to the levels in WT mice (Figure 1A). The residual *Nfe2l2* mRNA expression most likely originates from resident macrophages in the liver.

### 3.1. Experiment 1

Time-dependent changes in EE were compared in WT and *NRF2* KO mice prior to and after initial introduction of the MR diet. During the initial two days when all mice were consuming the Con diet, the cycling of changes in daytime and nighttime EE were comparable between WT and *NRF2* KO mice (Figure 1B,C). After introduction of the MR diet the daytime and nighttime EE continued to be similar among the groups for 2 more days, but between days 3 and 4, a larger increase in nighttime EE in both WT and *NRF2* KO mice on the MR diet became evident (Figure 1B,C). During the subsequent 5 days (e.g., 5–9), both daytime and nighttime EE in WT and *NRF2* KO mice on the MR diet were increased compared to mice of each genotype on the Con diet (Figure 1B,C). To assess the timing of diet-dependent changes in EE after introduction of dietary MR in more detail, the average nighttime EE during the 2 day run-in period was subtracted from the nighttime means of each group after MR was introduced and plotted as diet-dependent changes in nighttime EE from days 1 through 9. Figure 1D shows that nighttime EE in the WT and *NRF2* KO mice on the Con diet was essentially unchanged over this period. Nighttime EE in the WT and *NRF2* KO mice on the MR diet was also similar to mice on the Con diet from days 1 to 2, but on day 3, nighttime EE became significantly higher in WT and *NRF2* KO mice on the MR diet compared to their respective controls (Figure 1D). From days 3 to 6, the nighttime means of EE in WT and *NRF2* KO mice on the MR diet continued to increase, while EE in the corresponding control groups remained stable over this period (Figure 1D). Although not shown, daytime EE followed a similar pattern of change among the four groups over the 9 day period. EE was also measured after eight weeks of dietary MR and as shown in Figure 1E, the MR diet produced comparable increases in total EE in WT and *NRF2* KO mice. EE of *NRF2* KO mice on the Con diet was slightly higher than WT mice on the Con diet, but the MR diet produced a similar increment of increase in EE in both genotypes (Figure 1E). Collectively, the findings show that the time-dependent increases in EE produced after initial introduction of dietary MR or after eight weeks of dietary MR are mostly unaffected by the absence of liver *Nfe2l2*.

### 3.2. Experiment 2

The initial body weights and composition of WT and *NRF2* KO mice were comparable, and both genotypes responded to dietary MR with a similar decrease in accumulation of body weight and adiposity over the following weeks (Figure 2A,B). The MR diet also produced comparable increases in energy (Figure 2C) and water intake (Figure 2D) in both genotypes. Serum FGF21 levels in mice on the Con diet did not differ between WT and *NRF2* KO mice, and MR increased serum levels to ~15 ng/mL in both genotypes (Figure 2E). However, because basal levels were slightly lower in *NRF2* KO compared to WT mice on the Con diet, the fold induction by MR was higher in the *NRF2* KO compared to WT mice (Figure 2E). Given that the ability of dietary MR to affect parameters of energy balance are thought to be entirely FGF21-dependent [12], it is not surprising that the absence of *Nfe2l2* did not compromise the ability of dietary MR to increase EE and energy intake, while reducing body weight and adiposity.

### 3.3. Real Time qPCR

*Nfe2l2* is viewed as a master regulator of the transcriptional response to oxidative stress, and its activation induces a comprehensive program of redox target genes (reviewed in [46]). To establish whether *Nfe2l2* plays an essential role in mediating hepatic transcriptional responses to MR, a combination of targeted and non-targeted transcriptomic analysis was employed. Previous bioinformatic analysis of the hepatic responses to MR identified a number of *Nfe2l2* target genes and ER stress-response genes that were up- or down-regulated in a manner consistent with *Nfe2l2* activation. qPCR analysis of a sampling of this gene set revealed complex, unanticipated effects of *Nfe2l2* deletion on their expression (Table 1). For example, basal expression of four *Nfe2l2* target genes (*aldehyde oxidase 1 [Aox1], glutathione-disulfide reductase [Gsr], thioredoxin reductase [Txnrd]1, glutamate-cysteine ligase catalytic subunit [Gclc]*) and three ER stress genes (*tribbles homolog 3 [Trb3], spliced X-Box-binding protein 1 [Xbp1s]*, and *Atf4*) was suppressed in *NRF2* KO mice on the Con diet, but the expression of these genes was unaffected by MR regardless of genotype (Table 1, highlighted in green). A second subset of genes also required *Nfe2l2* to maintain basal expression in mice on the Con diet (e.g., *epoxide hydrolase 1 [Ephx1]*, *superoxide dismutase 2 [Sod2], and glutathione-S-transferase alpha 2 [Gsta2]*), and the absence of *Nfe2l2* prevented their induction or repression by dietary MR (Table 1, highlighted in red). In a third subset of genes (e.g., *carbonyl reductase 1 [Cbr1], microsomal glutathione S transferase [Mgst3]*, and *NADPH quinone dehydrogenase 1 [Nqo1])*, basal expression was unaltered by the absence of *Nfe2l2*, but its absence did compromise the induction of these genes by MR in *NRF2* KO mice (Table 1, highlighted in blue). Basal expression was also unaffected by the absence of *Nfe2l2 in* a fourth subset of ER stress genes (e.g., *asparagine synthetase [Asns]*, *cytochrome P450 family 4 subfamily a polypeptide 14 [Cyp4a14]*, *Fgf21*, *phosphoserine aminotransferase 1 [Psat1], very low density lipoprotein receptor [Vldlr]*), and their induction by dietary MR was fully intact (Table 1, highlighted in yellow). Together, these findings suggest that dietary MR does not act singularly through *Nfe2l2* to mediate its transcriptional effects on genes collectively involved in the integrated stress response, and that a more complex regulatory model is needed to explain the transcriptional adaptation to reduced dietary methionine.

### 3.4. Differential Gene Expression in the Liver

To obtain a more in-depth and unbiased assessment of the respective responses of the two genotypes to dietary MR, we used RNAseq in conjunction with bioinformatic analysis to interrogate the transcriptional responses in livers from WT and *NRF2* KO mice fed the Con and MR diets. A comparison of gene expression within genotype showed a statistically significant overlap of differentially expressed genes between the two genotypes. Using a pairwise comparison of 5286 genes with adjP ≤ 0.1 for log fold change of WT MR versus WT Con and *NRF2* KO MR versus *NRF2* KO Con, the correlation of the effect of MR in the two genotypes was highly significant (R^2^ = 0.64, *p* < 2.2 × 10^−16^, Appendix A).

To further explore the systems biology of hepatic transcriptional responses to MR in the respective genotypes, Ingenuity Pathway Analysis (IPA) was used to screen differentially expressed genes against Ingenuity’s Knowledge Base annotated database to detect predicted activation or inhibition of canonical pathways and upstream regulators. This algorithm predicts changes in canonical pathways and transcription factor activity based on observed patterns of change in expression of genes known to be up- or down-regulated by specific transcription factors. Heat maps were constructed to illustrate the comparative effects of the MR diet in the two genotypes on the twenty most up- or down-regulated canonical pathways (Figure 3A) and transcription factors (Figure 3B). When canonical biological pathways were evaluated in terms of activation or inhibition, a similar conclusion was reached in terms of the relative effects of MR in the two genotypes (Figure 3A). For example, canonical pathways that were comparably activated across genotype include cholesterol biosynthesis, eIF2α signaling, *Nfe2l2*-mediated oxidative stress responses, xenobiotic metabolism, RhoGDI signaling, tRNA charging, and glutathione-mediated detoxification. Inhibited pathways, which were mostly immune related, were slightly more repressed in livers from WT mice (Figure 3A). Overall, the directionality of the effects of MR between genotypes was consistent across all canonical pathways, and the only differences noted were in the strength of activation or inhibition of some pathways (Figure 3A). Comparison of the top upstream regulators inhibited by MR in WT and *NRF2* KO mice shows that the translational regulator LARP, an RNA-binding protein that regulates translation of target mRNA species downstream of the mTORC1 complex, and Rictor (mTORC1 component) are strongly downregulated by MR in both genotypes (Figure 3B). TNF, interferon gamma, and other genes associated with inflammatory signaling are also comparably inhibited by MR in both genotypes (Figure 3B). Transcription factors and upstream regulators that were strongly activated by MR in both genotypes included MYCN, MYC, the transcriptional repressor MLXIPL, the mitochondrial mRNA processing protein PNPT1, and the transcriptional co-activator TRIM24. Of particular note is that IPA detected a comparable activation of *Nfe2l2* in WT and *NRF2* KO mice (Figure 3B). This finding suggests that many of the predicted *Nfe2l2* target genes receive multiple additional regulatory inputs in the context of the MR diet. When viewed collectively over a range of canonical pathways and upstream regulators, the responses to the MR diet in WT and *NRF2* KO mice were convincingly comparable (Figure 3A,B).

To explore the impact of genotype and diet on the transcriptome more fully, we used GSEA to identify gene sets that are enriched across genotype and KEGG pathways that are activated or inhibited by MR in both genotypes. For example, the GSEA enrichment plots and associated heat maps for the KEGG Drug Metabolism Cytochrome P450 pathway presented in Figure 4A,B show that dietary MR produces a comprehensive activation of components of this pathway in both WT and *NRF2* KO mice. An MR-dependent increase in xenobiotic metabolism was detected by IPA in both WT and *NRF2* KO mice as well (Figure 3A). This is likely due to the substantial overlap between these pathways across platforms. On its surface these findings suggest that the absence of *Nfe2l2* had little impact on activation of these pathways by MR. However, heat maps of the KEGG Drug Metabolism pathway show that the genes in this pathway are more highly expressed in WT versus *NRF2* KO mice on both the MR diet (Figure 4C) and the Con diet (Figure 4D). Together, these findings suggest that *Nfe2l2* may be important in maintaining basal expression of genes within this pathway but is not essential in mediating the MR-dependent increase in their expression. This is reminiscent of our findings for a subgroup of *Nfe2l2*-sensitive genes in Table 1, where the absence of *Nfe2l2* impacted their basal expression but not their induction.

The KEGG Ribosome pathway is another example where GSEA detected a significant impact of the absence of *Nfe2l2* in the response to MR, but in this case basal expression of genes within this pathway were unaffected in *NRF2* KO mice compared to WT mice. Dietary MR produced a strong induction of the KEGG Ribosome pathway in WT mice (Figure 5A) whereas MR was relatively ineffective in *NRF2* KO mice (Figure 5B). These findings point to a previously unappreciated role for *Nfe2l2* in mediating the MR-dependent effects on the KEGG Ribosome pathway. The KEGG Pentose Glucuronate Interconversions pathway is another example where the absence of hepatic *Nfe2l2* had little effect on expression of genes within this pathway in mice on the Con diet, but MR produced essentially no induction of pathway genes in *NRF2* KO mice (Figure 5C). In contrast, MR produces a robust induction of the KEGG Pentose Glucuronate Interconversions pathway in WT mice (Figure 5C). Lastly, GSEA identified the KEGG SREBP2 pathway as being equivalently induced by MR in both genotypes (Appendix A), while genotype had no effect on expression of genes within this pathway. Given the role of SREBP2 in mediating expression of genes involved in cholesterol synthesis, these findings are fully consistent with Figure 3A showing robust induction of four different canonical pathways of cholesterol biosynthesis in both genotypes. Collectively, these findings make a compelling case that although *Nfe2l2* plays a complex, multifunctional role as a mediator of diverse transcriptional responses to dietary MR in the liver, their absence in *NRF2* KO mice did not compromise the transcriptional responses that link dietary MR to biological effects on energy balance and lipid metabolism. We conclude that *Nfe2l2* is not an essential link between dietary MR and its metabolic phenotype.

## 4. Discussion

Within six hours of introducing a diet low in methionine (e.g., 0.17%) to mice, the diet produces a coordinated activation of transcriptional programs in the liver that function to compensate for the reduced intake of methionine. These short-term adaptive responses increase hepatic amino acid levels through immediate changes in RNA biology, increased tRNA aminoacylation, decreased ternary complex formation, and an overall decrease in protein translation [47]. In the days that follow, these initial compensatory responses limit the decrease in hepatic methionine and glutathione to 50% of control levels despite the five-fold decrease in dietary methionine concentration. The compensatory responses are also able to maintain redox homeostasis in the context of reduced hepatic glutathione. Another key target of the early transcriptional response to MR is hepatic FGF21, and the corresponding increase in circulating FGF21 can be detected within hours after introduction of the MR diet [7]. The elevated FGF21 produces a coordinated increase in energy intake and energy expenditure (EE) that limits fat deposition over time and produces a systematic improvement in biomarkers of metabolic health [12]. A key unresolved question in the field is how the sensing of methionine in the liver is translated into the transcriptional activation of hepatic FGF21.

Significant effort has been directed towards identifying the nutrient-sensing molecules and transcription factors that link dietary MR to the components of its metabolic phenotype, but the anatomical organization and molecular identity of these translational mechanisms are poorly understood. A prime candidate is hepatic ATF4. *Atf4* mRNA is preferentially translated after essential amino acid restriction, and the encoded protein (e.g., ATF4) is a master regulator of the ISR [48,49,50]. ATF4 can also heterodimerize with additional transcription factors to target the promoters of an expanded group of genes that provide protection against endogenous oxidants and electrophiles in the liver. This group includes enzymes such as cytochrome P450, glutathione S-transferases and UDP-glucuronosyl transferases that conjugate the products of cytochrome P450s with hydrophilic groups to facilitate their excretion, antioxidant enzymes like superoxide dismutases, glutathione peroxidase, catalase, and thiol containing molecules such as glutathione and thioredoxin that maintain reducing conditions within the cell and inactivate electrophilic compounds [51]. Many of these enzymes are encoded by genes containing antioxidant response elements (AREs) in their promoters. AREs are cis-acting enhancers that transcriptionally activate genes in response to changes in cellular redox status. Previous work has shown that dietary MR is also an effective activator of genes with AREs in their promoters [25]. The ability of *Nfe2l2* to heterodimerize with ATF4 expands the ISR transcriptional program to include additional redox genes [28,29], raising the interesting possibility that hepatic *Nfe2l2* could be an important contributor to the transcriptional responses to dietary MR.

Under normal conditions, the protein encoded by *Nfe2l2* (e.g., NRF2) is mainly localized in the cytoplasm through an interaction with Kelch-like ECH-associated protein 1 (KEAP1) and the actin cytoskeleton. Although *Nfe2l2* mRNA is constitutively expressed, KEAP1 targets NRF2 for polyubiquitination and degradation, effectively limiting the half-life of the protein. During exposure to electrophiles or oxidative stress, KEAP1 becomes oxidized at critical cysteine residues, releasing NRF2 to translocate to the nucleus and promote the expression of ARE-containing genes [49,52]. In the case of dietary MR, previous work has shown that the restriction of dietary methionine limits the formation of hepatic GSH and activates PERK through a mechanism independent of ER stress [25]. PERK is not typically activated by EAA depletion [30] so it seems likely that the ability of MR to activate PERK and initiate components of the ISR derives from its unique effects on hepatic GSH. Although NRF2 can be activated by PERK [31], NRF2 transcriptional activity also responds to low GSH [37]. Based on these findings, it seems logical that the ISR and the antioxidant stress response share genes whose primary function is to increase production of GSH [53]. Given the inferred role for hepatic NRF2 as a mediator of a significant number of MR-dependent transcriptional components of the ISR [25], the present work employed a loss of function approach to assess whether hepatic NRF2 is an essential link between those previously established transcriptional targets of dietary MR and the short-term metabolic responses to dietary MR. Based on our comprehensive phenotyping of the behavioral, physiological, endocrine, biochemical, and transcriptional endpoints herein, the answer to the larger question is that NRF2 is not an essential mediator of the near-term metabolic effects of dietary MR.

The findings of the present work documented many transcriptional programs that were mostly unaltered by the absence of hepatic *Nfe2l2*, but in Figure 4 and Figure 5, GSEA documented three different pathways that were altered by genotype irrespective of diet (Figure 4) or differed between genotypes in their response to dietary MR (Figure 5). For example, the absence of *Nfe2l2* produced a coordinated downregulation of genes in the KEGG Drug Metabolism Cytochrome P450 family in mice on the Con diet, but its absence also blunted the induction of genes in this pathway by MR in the *NRF2* KO mice relative to WT mice (Figure 4C). In contrast, genes within the KEGG Ribosome pathway were comparably expressed between WT and *NRF2* KO mice on the Con diet, but were much less robustly induced by MR in the *NRF2* KO group (Figure 5A,B). Additionally, in the KEGG Pentose Glucuronate Interconversions pathway, which provides glucuronides for conjugation to enhance excretion of toxic compounds, MR was much less effective in inducing genes of this pathway in the *NRF2* KO group (Figure 5C). Given the protective role of each pathway, an important question is whether the absence of *Nfe2l2* in the liver would compromise the longer-term responses to dietary MR. The initial work on dietary MR in rats showed that the diet increased longevity [1,2], while also producing chronic improvements in glucose homeostasis and reductions in adiposity [4]. Subsequent work showed that MR also produced long term reductions in markers of inflammation in both adipose tissue and the liver [10]. The study by Wanders et al. [10] provided a direct, age-matched comparison of the relative efficacy of MR versus calorie restriction, and MR was significantly more effective in reducing expression of inflammatory markers in both sites. The relevance of this point comes into focus from a recent assessment of the ability of 30% calorie restriction to increase longevity in mice with a global deletion of *Nfe2l2* [54]. The authors found that global *NRF2* KO mice fed ad libitum were shorter lived than wild type mice, but surprisingly, calorie restriction produced a comparable increase in longevity between wild type and *NRF2* KO mice [54]. Based on the current findings showing that the absence of hepatic *Nfe2l2* compromised the MR-dependent induction of drug metabolism/detox pathways in the liver, it remains an open question of whether liver-specific deletion of *Nfe2l2* would limit the life extending effects of dietary MR. Based on the compromised responses to MR identified here, the outcome might depend on whether the liver-specific *NRF2* KO animals were exposed to significant dietary or environmental stressors. At least in outbred mice and other longevity models, the absence of *Nfe2l2* compromises stress resistance and lifespan [55,56,57], although in mice, the life-shortening pathology originated in non-hepatic tissues. It will be interesting in future studies to examine how the absence of hepatic *Nfe2l2* impacts the longer-term effects of dietary MR.

## 5. Conclusions

The present work establishes that *Nfe2l2* is not an indispensable mediator of the short-term, metabolic effects of dietary MR.

## Figures and Tables

**Figure 1 nutrients-13-01788-f001:**
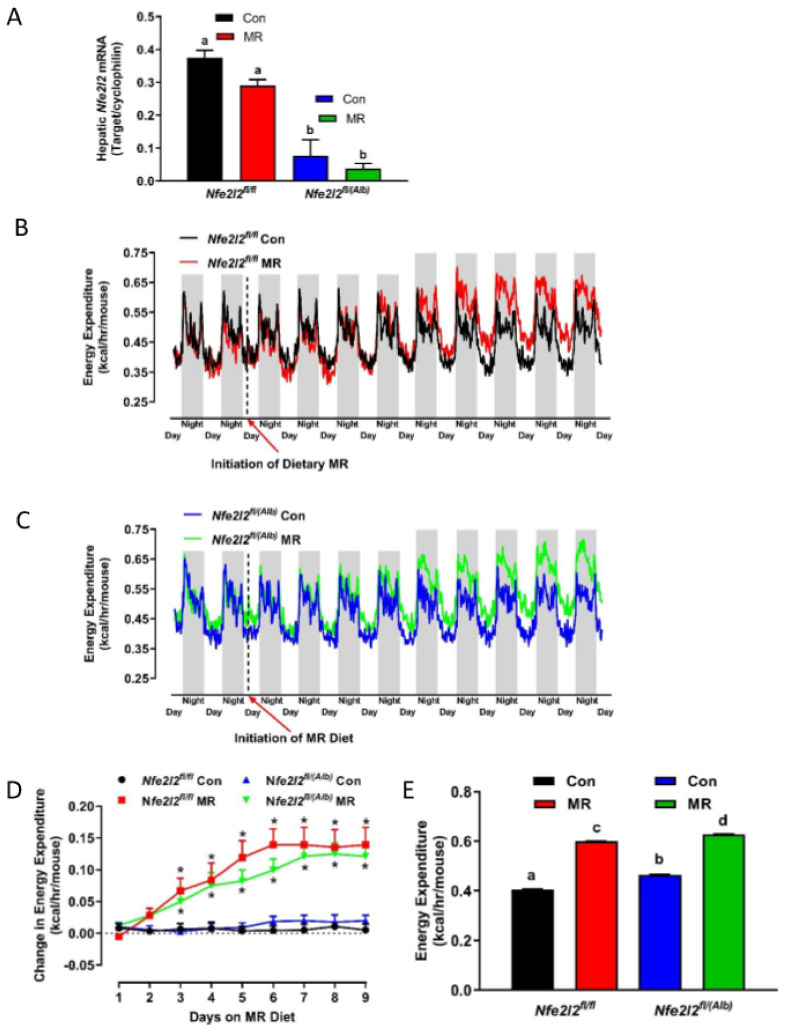
Hepatic *Nfe2l2* mRNA (**A**), initial changes in energy expenditure (EE) after introduction of dietary MR (**B**–**D**), and change in 24 h EE at the end of the experiment (**E**) in *Nfe2l2^fl/fl^* (WT) and *Nfe2l2^fl/(Alb)^* (*NRF2* KO) mice fed Con or MR diets. Primers designed to detect excised sequences in exon 5 of the *Nfe2l2* gene were used to detect *Nfe2l2* mRNA in livers from mice of the two genotypes by real time qPCR (**A**). EE was measured and analyzed during the run-in period and at the end of the experiment by ANCOVA as described in the Materials and Methods (**B**–**E**). In (**D**), the initial diet-dependent change in nighttime EE was calculated by subtracting the nighttime EE of each group prior to introduction of MR from the nighttime measures of EE during the 9 days after initial exposure to the MR diet on day 0. Means are representative of data from 7 or 8 mice per group and compared by two-way ANOVA as described in the Materials and Methods. Means annotated with a different letter differ at *p* < 0.05.

**Figure 2 nutrients-13-01788-f002:**
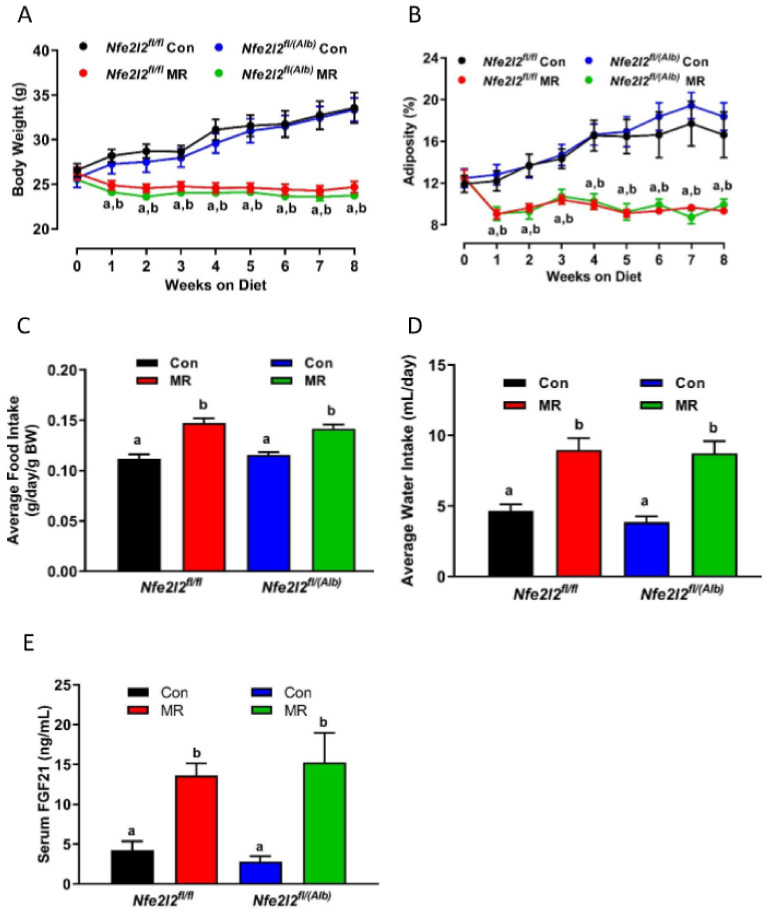
Assessment of impact of dietary MR on body weight (**A**), adiposity (**B**), average food intake (**C**), average water intake (**D**), and serum FGF21 (**E**) in *Nfe2l2^fl/fl^* (WT) and *Nfe2l2^fl/(Alb)^* (*NRF2* KO) mice fed Con or MR diets for 8 weeks beginning at 8 weeks of age. Body weight, adiposity, food intake, and water intake were measured weekly for the entire study. The means of food intake and water intake were averaged over the entire study, while serum collected at the end of the study was used for measurement of FGF21. The change in body weight and adiposity over time were analyzed using a repeated measures two-way ANOVA as described in the Materials and Methods, and the Con means annotated with an ‘a’ differ between *Nfe2l2^fl/fl^* mice fed Con or MR diets, and Con means annotated with a ‘b’ differ between *Nfe2l2^fl/(Alb)^* mice fed Con or MR diets. Food intake, water intake, and serum FG21 were analyzed by two-way ANOVA and means annotated with different letters differ at *p* < 0.05. Data in each figure panel are presented as the mean ± SEM, *n* = 7–8.

**Figure 3 nutrients-13-01788-f003:**
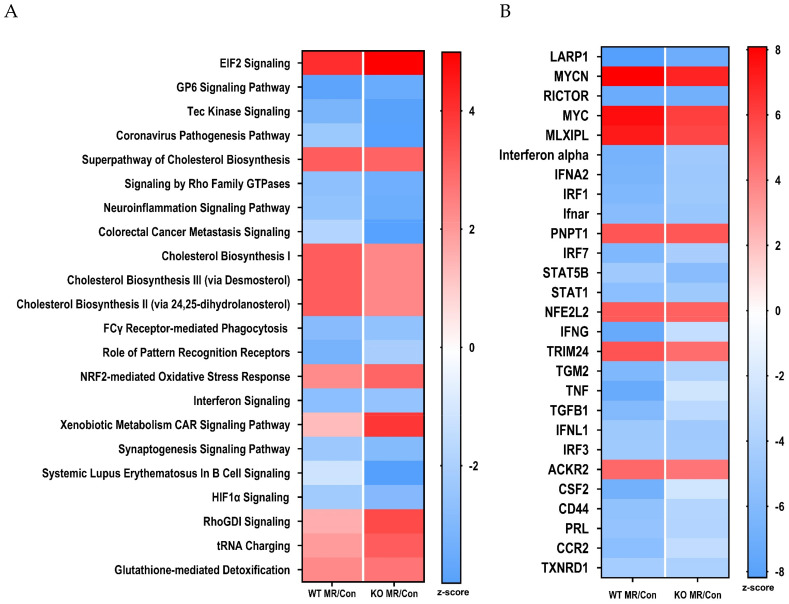
Bioinformatics analysis of hepatic gene expression in *Nfe2l2^fl/fl^* (WT) and *Nfe2l2^fl/(Alb)^* (*NRF2* KO) mice fed Con or MR diets for 8 weeks. Canonical pathway analysis (**A**) and upstream regulator analysis (**B**) were conducted using Ingenuity Pathway Analysis as described in the Materials and Methods. Pathway analysis was conducted using livers from 5 to 6 mice of each genotype × diet combination and utilized to identify putative gene regulators responsible for the observed transcriptional patterns produced by the MR diet compared to the Con diet in the two genotypes. Upstream regulators and canonical pathways with an activation z-score ≥2 or ≤2 were considered to be activated (red) or inhibited (blue), respectively. Heat maps were used to visualize the top 20 canonical pathways and upstream regulators that were differentially affected by the MR diet in the two genotypes.

**Figure 4 nutrients-13-01788-f004:**
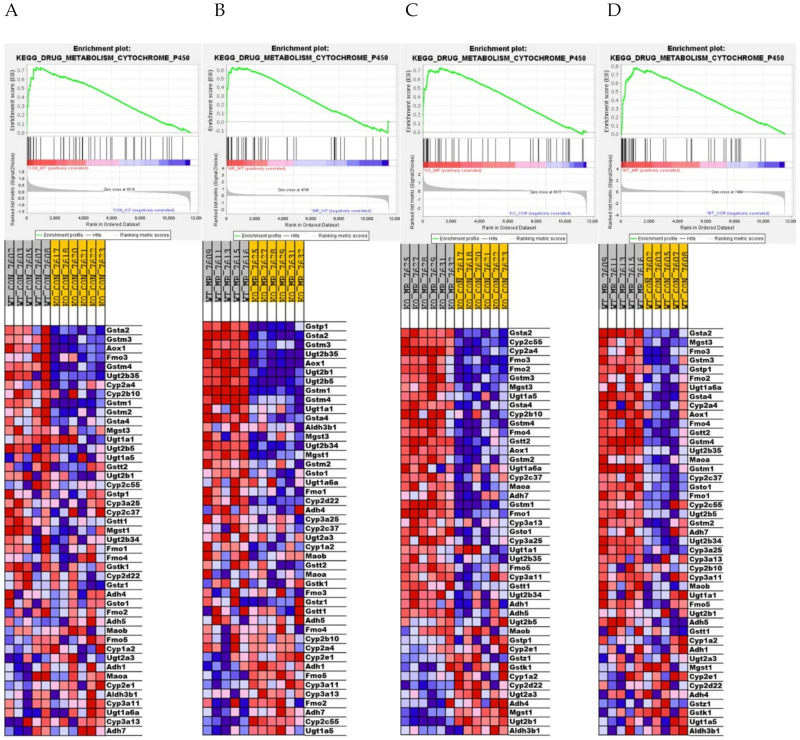
Gene set enrichment analysis (GSEA) of hepatic gene expression in *Nfe2l2^fl/fl^* (WT) and *Nfe2l2^fl/(Alb)^* (*NRF2* KO) mice showing the differential effect of genotype and diet on the top-scoring KEGG Drug Metabolism Cytochrome P450 gene set. Enrichment was computed as described in the Material and Methods and the individual contributions of pathway genes to the pathway enrichment signal were visualized via enrichment plots depicting the trajectory of a normalized pathway enrichment score against the rank of the pathway genes in the context of the full gene list. Accompanying heat maps present the normalized enrichment scores for individual genes within the gene set [blue = downregulation, red = upregulation, gray = not significant (FDR > 0.1)]. Panel (**A**) shows the enrichment set for the WT MR to WT Con comparison, panel (**B**) shows the enrichment set for the *NRF2* KO MR to *NRF2* KO Con comparison, panel (**C**) shows the enrichment set for the WT MR to *NRF2* KO MR comparison, and panel (**D**) shows the enrichment set for the WT Con to *NRF2* KO Con comparison.

**Figure 5 nutrients-13-01788-f005:**
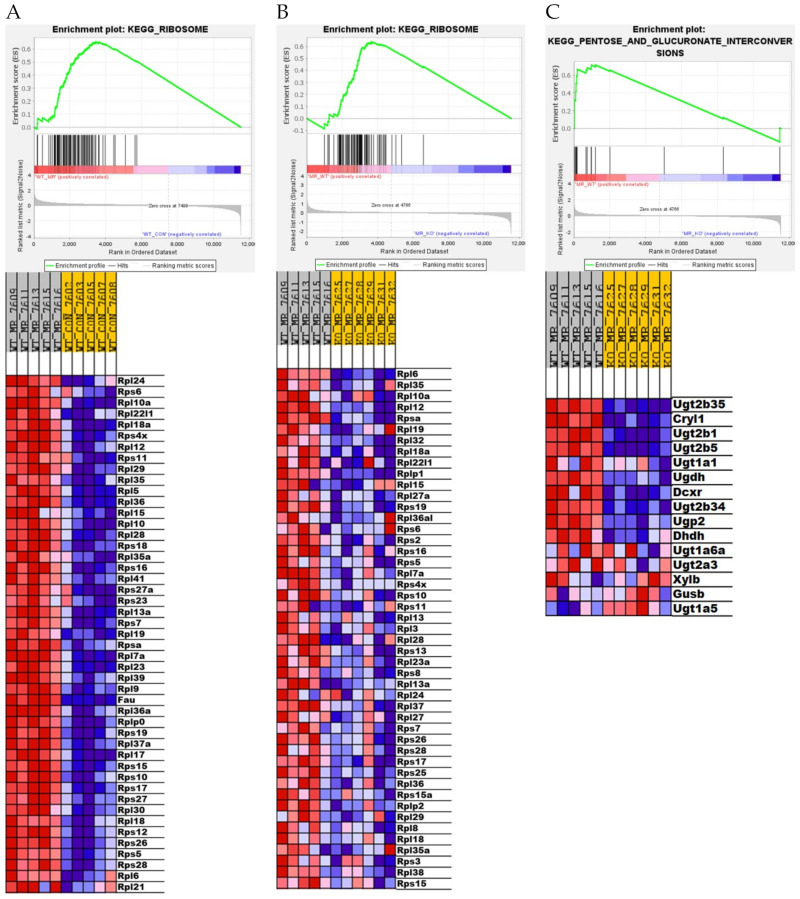
Gene set enrichment analysis (GSEA) of hepatic gene expression in *Nfe2l2^fl/fl^* (WT) and *Nfe2l2^fl/(Alb)^* (*NRF2* KO) mice showing the differential effect of genotype and diet on the top-scoring KEGG Ribosome Drug gene set (**A**,**B**) and KEGG Pentose and Glucuronate Interconversions gene set (**C**). Enrichment was computed as described in the Material and Methods and the individual contributions of pathway genes to the pathway enrichment signal were visualized via enrichment plots depicting the trajectory of a normalized pathway enrichment score against the rank of the pathway genes in the context of the full gene list. Accompanying heat maps present the normalized enrichment scores for individual genes within the gene set [blue = downregulation, red = upregulation, and gray = not significant (FDR > 0.1)]. Panel (**A**) shows the enrichment set for Ribosome pathway for the WT MR to WT Con comparison, panel (**B**) shows the enrichment set for the Ribosome pathway for the WT MR to *NRF2* KO MR comparison, while panel (**C**) shows the Pentose and Glucuronate Interconversions enrichment set for the WT MR to *NRF2* KO MR comparison.

**Table 1 nutrients-13-01788-t001:** Genetic markers of NRF2 transcriptional program and ER stress response in livers of *Nfe2l2^fl/fl^* and *Nfe2l2^fl/(Alb)^* mice fed Control or Methionine-restricted (MR) diets for 8 weeks.

Gene	Signaling	*Nfe2l2^fl/fl^*		*Nfe2l2^fl/(Alb)^*	
Symbol ^1^	Pathway	Control	MR	Control	MR
***Aox1***	Nfe2l2	1 ± 0.25 ^a^	1.28 ± 0.20 ^a^	0.35 ± 0.16 ^b^	0.40 ± 0.09 ^b^
***Gclc***	Nfe2l2	1 ± 0.13 ^a^	0.90 ± 0.07 ^a^	0.45 ± 0.04 ^b^	0.38 ± 0.03 ^b^
***Gsr***	Nfe2l2	1 ± 0.08 ^a,b^	1.20 ± 0.11 ^a^	0.73 ± 0.05 ^b^	0.86 ± 0.12 ^b^
***Txnrd1***	Nfe2l2	1 ± 0.04 ^a^	1.20 ± 0.18 ^a^	0.68 ± 0.05 ^b^	0.80 ± 0.05 ^a,b^
***Trb3***	ER stress	1 ± 0.22 ^a^	1.24 ± 0.13 ^a^	0.37 ± 0.05 ^b^	0.71 ± 0.08 ^a,b^
***Atf4***	ER Stress	1 ± 0.05 ^a,b^	1.11 ± 0.08 ^b^	0.68 ± 0.04 ^c^	0.86 ± 0.04 ^a,c^
***Xbp1s***	ER Stress	1 ± 0.19 ^a^	0.66 ± 0.13 ^a,b^	0.45 ± 0.06 ^b,c^	0.44 ± 0.07 ^b,c^
***Ephx1***	Nfe2l2	1 ± 0.17 ^a^	1.42 ± 0.07 ^b^	0.35 ± 0.11 ^c^	0.43 ± 0.05 ^c^
***Sod2***	Nfe2l2	1 ± 0.18 ^a^	0.60 ± 0.08 ^b^	0.54 ± 0.04 ^b^	0.62 ± 0.07 ^b^
***Gsta2***	Nfe2l2	1 ± 0.32 ^a^	4.86 ± 0.63 ^b^	0.15 ± 0.09 ^c^	0.66 ± 0.20 ^a,c^
***Cbr1***	Nfe2l2	1 ± 0.20 ^a^	3.45 ± 0.52 ^b^	0.93 ± 0.14 ^a^	1.10 ± 0.17 ^a^
***Mgst3***	Nfe2l2	1 ± 0.13 ^a^	1.97 ± 0.16 ^b^	0.92 ± 0.18 ^a^	1.34 ± 0.18 ^a^
***Nqo1***	Nfe2l2	1 ± 0.15 ^a^	2.28 ± 0.16 ^b^	0.39 ± 0.09 ^a^	0.71 ± 0.13 ^a^
***Asns***	ER stress	1 ± 0.32 ^a^	12.23 ± 1.49 ^b^	0.74 ± 0.16 ^a^	10.59 ± 1.59 ^b^
***Cyp4a14***	ER stress	1 ± 0.21 ^a^	1.74 ± 0.23 ^b^	0.97 ± 0.25 ^a^	2.13 ± 0.19 ^b^
***Fgf21***	ER stress	1 ± 0.36 ^a^	2.77 ± 0.60 ^b^	1.04 ± 0.20 ^a^	2.90 ± 0.76 ^b^
***Psat1***	ER stress	1 ± 0.33 ^a^	7.54 ± 0.96 ^b^	0.48 ± 0.12 ^a^	3.27 ± 0.54 ^c^
***Vldlr***	ER stress	1 ± 0.21 ^a^	3.25 ± 0.76 ^b^	1.02 ± 0.10 ^a^	3.79 ± 0.59 ^b^

^1^ Messenger RNA expression of hepatic markers of *Nef2l2* transcriptional program (*Aox1*, *Cbr1*, *Ephx1*, *Gclc*, *Gsr*, *Gsta2*, *Mgst3*, *Nqo1*, *Sod2*, *Trib3*, *Txnrd1*) and ER stress (*Atf4, Xbp1s, Asns, Cyp4a14, Fgf21, Psat1, Vldlr*) in livers of *Nfe2l2^fl/fl^* and *Nfe2l2^fl/(Alb)^* mice (*n* = 8 per group) were determined by RT-PCR, expressed relative to cyclophilin, adjusted to fold induction from *Nfe2l2^fl/fl^* Control, and compared by two-way ANOVA to test for effects of genotype, diet, and genotype × diet interaction. Residual variance was used as the error term for post hoc testing of genotype × diet means for each gene using the Bonferroni correction. Within each gene, means annotated with different letters differ at *p* < 0.05. Gene symbols highlighted in green are those whose basal expression was reduced by deletion of *Nfe2l2*. Gene symbols highlighted in red were those that *Nfe2l2* deletion lowered basal expression and blocked MR-dependent induction. Basal expression of genes highlighted in blue were unaffected by *Nfe2l2* deletion but their MR-dependent induction was blocked, while expression of genes highlighted with yellow were unaffected by deletion of *Nfe2l2*.

## Data Availability

All data generated or analyzed during this study will be available from the lead contact upon request.

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
