# Peer review of "Hepatic Nfe2l2 Is Not an Essential Mediator of the Metabolic Phenotype Produced by Dietary Methionine Restriction"

_nutrients, 2021, doi:10.3390/nu13061788_

Round 1

Reviewer 1 Report

This is a part of consecutive studies published from the identical research group concerning methionine restriction. Although the results were not as they had expected, the present investigation was designed appropriately and conducted by rigid methodology. I do not have any concerns for this manuscript. I'm looking forward to seeing their next article of the same theme.

Author Response

Response to Reviewer 1's comments are provided in attached file.

Reviewer 2 Report

This paper is very well written and conclusions are supported by the mentioned literature. In general, it is easy to follow, but before publishing, I only have minor suggestions:

Introduction:

In regard to the mentioned eIF2alpha kinases (GCN2, PERK..) and ATF4 (lines 64-74) it is of interest to describe a gender specificity (if possible).

Methods:

What is the rationale using young (6 weeks old) mice?

Results:

Table 1 is described very well in the manuscript but was not included in the manuscript?!

Line 309: remove typo “KO”

Author Response

Response to Reviewer 2's comments are provided in attached file.

Reviewer 3 Report

The manuscript describes in detail biological consequences of an absence of hepatic NRF2 (product of Nfe212), which is thought to play a significant role in transcriptional response related to dietary methionine restriction. Using WT and hepatic NRF2 KO mice fed with normal or methionine-restricted diet, authors determined by combining phenotypic characterization of mice (e.g. body weight, fat content, energy expenditure) and their hepatic gene expression that hepatic NRF2 had no effect on the ability of the restricted methionine intake to mediate the metabolic effects attributed to the methionine restriction. In addition, gene expression was similarly affected in WT and KO mice on low-methionine diet with exception of P450-related drug metabolism and pentose-glucuronate interconversion pathways. Overall, authors concluded that hepatic NRF2 is not essential in mediating the metabolic effects of dietary methionine restriction.

Manuscript is well written, straightforward and easy to follow. I have just a couple of comments for authors to respond and implement the potential correction into the revised manuscript:

  • Why did you decide to use only male mice? Is it reasonable to expect that deletion of Nfe212 would have different consequences in females? It would be interesting to see males and females data side by side.
  • In Methods (line 159) you specify using 10 ng of cDNA for assay. However, it is not clear whether these 10 ng was actual determined value or somehow assumed from the input of 2 ug of total RNA into reverse transcription assay. Please clarify.
  • Figure 1D and lines 252-270: This nighttime versus daytime energy expenditure comparison is interesting, but at the same time confusing (at least for me). Figure 1D shows points for the days, but text focuses on nighttime. Is it because mice are mostly active at night?
  • Line 305: I was unable to find Table 1 anywhere in the pdf version of the manuscript I reviewed.
  • Line 309: remove extra “KO”
  • Figures 4 and 5: These are relatively large figures, which can hardly fit on a single page (each). Perhaps, only heatmaps with listed genes are sufficient to show in the manuscript and rest could be in the supplement. This way each figure could fit on a single page.
  • Supplement: Legends for the supplementary figures should be with the respective figures and not in the footnote of the manuscript. Ideally, a single page would show an entire figure along with its legend.

Author Response

Responses to Reviewer 3's comments are provided in attached file.
